

# Comparison of the composition and function of gut microbes between adult and juvenile *Cipangopaludina chinensis* in the rice snail system

Kangqi Zhou[1,*], Junqi Qin[1,*], Haifeng Pang[2], Zhong Chen[1], Yin Huang[1], Wenhong Li[2], Xuesong Du[1], Luting Wen[1], Xianhui Pan[1] and Yong Lin[1]

[1] Guangxi Key Laboratory for Aquatic Genetic Breeding and Healthy Aquaculture, Guangxi Academy of Fishery Sciences, Nanning, China
[2] Guangxi University, Nanning, China
* These authors contributed equally to this work.

Corresponding authors
Xianhui Pan, 15002381261@163.com
Yong Lin, linnn2005@126.com

## ABSTRACT

*Cipangopaludina chinensis* is an important economic value snail species with high medicinal value. The gut microbes of aquatic animals plays a vital role in food digestion and nutrient absorption. Herein, we aimed at high-throughput sequencing of 16S rRNA to further investigate whether there were differences in the composition and function of gut microbes of adult and juvenile *C. chinensis* snails, as well as sediments. This study found that the microbial diversity of the sediment was significantly higher than that of the snails gut ($P < 0.001$), but there was no significant difference between the gut flora of adult and juvenile snails ($P > 0.05$). A total of 47 phyla and 644 genera were identified from all samples. Proteobacteria and Verrucomicrobia were the two dominant phyla in all samples, and overall relative abundances was 48.2% and 14.2%, respectively. Moreover, the relative abundances of *Aeromonas* and *Luteolibacter* in the gut of juvenile snails (30.8%, 11.8%) were higher than those of adults (27.7%, 10.6%) at the genus level ($P > 0.05$). Then, four indicator genera were found, namely *Flavobacterium*, *Silanimonas*, *Geobacter* and *Zavarzinella*, and they abundance in the gut of juvenile snails was significantly higher than that of adults ($P < 0.05$). This results imply the potential development of *Silanimonas* as a bait for juvenile snail openings. We observed that *Aeromonas* was the primary biomarker of the snail gut and sediments ($P < 0.001$), and it may be a cellulose-degrading bacteria. Function prediction revealed significantly better biochemical function in the snail gut than sediments ($P < 0.001$), but no significant differences in adult and juvenile snail ($P > 0.05$). In conclusion, studies show that the snail gut and sediment microbial composition differ, but the two were very similar. The microbial composition of the snail gut was relatively stable and has similar biological functions. These findings provide valuable information for in-depth understanding of the relationship between snails and environmental microorganisms.

## INTRODUCTION

*Cipangopaludina chinensis* is one of the most common large freshwater snails and is widely distributed in Asian countries (*Lu et al., 2014*). This variety has delicious meat and high nutritional value (*e.g.*, protein exceeds 12%, fat is only about 0.6%, and it is rich in more than 40% of umami amino acids), and is favored by consumers and farmers in China (*Zhou et al., 2021*; *Luo et al., 2021*). The annual output value of using *C. chinensis* as a food material exceeds billions in the catering industry alone. In addition, studies have found that polysaccharides from *C. chinensis* (CCPS) has a variety of biological activities (*Xiong et al., 2016*; *Xiong et al., 2019*). In terms of anti-cancer, *Liu, Li & Wang (2013)* used the 2.2.15 cell line of human hepatoma cells (HepG2) cloned and transfected with HBV-DNA as an *in vitro* experimental model to prove that the CCPS has obvious anti-HBV effect. Then, *Zhu et al. (2016)* confirmed the anti-tumor effect of CCPS by using human cervical cancer cell line (Hela) and human colorectal cancer cell line (HCT-8) as *in vitro* experimental models. Moreover, studies have reported that it also plays an important role in heptoprotective. *Fan, Li & Wang (2014)* demonstrated through an alcohol-induced liver injury model that CCPS can reduce ALT, AST activity and MDA content, increase SOD activity, increase GSH content, revealing that it has a protective effect on alcohol-induced liver injury. The results of *Jiang et al. (2013)* showed that CCPS has a significant protective effect on BCG/LPS-induced immune liver injury through combined *in vitro* and *in vivo* experiments. Therefore, *C. chinensis* can not only be used as food, but also have great potential in human disease prevention and treatment.

Gut microbes have been proved to play an important role in nutrient absorption, physiological metabolism and immune defense, and are essential factors for maintaining the health of aquatic animals (*Mitev & Taleski, 2019*; *Yadav & Jha, 2019*). There may also be a similar effect in *C. chinensis*. Currently, the composition and function of the gut microbiota in insects, fish and mammals have been well studied, but the gut microbiota of snails has not been systematically and deeply studied. Additionally, we have observed differences in food preferences between juvenile and adult *C. chinensis* in actual production, which may be caused by differences in the gut microbiota (*Zhou et al., 2021*). Herein, it is critical to initially understand whether the development or growth of *C. chinensis* affects the gut microbial community composition and functions. 16S rRNA is ubiquitous in prokaryotic cells, and has the advantages of good stability, high sequence conservation, large amount of information, and easy extraction, so it is widely used as an ideal material for the study of animal intestinal flora (*Langille et al., 2013*; *Hu et al., 2018*; *Li et al., 2019*). In this study, we performed high-throughput sequencing of 16S rRNA gene to study the function and composition of intestinal microbiota in adult and juvenile *C. chinensis* under artificial habitat. This result study fills the gap in our understanding of the gut microbiota in different developmental stages of *C. chinensis* (adult and juvenile stage), and provides an important reference for the application potential of high-throughput technology. In this way, it can guide the production of the snail industry more scientifically and effectively, and provide scientific materials for promoting the development of snail commercial feed.

## MATERIALS AND METHODS

### Ethics statement

All animal experiments were conducted in accordance with the guidelines and approval of the Institutional Animal Care and Use Committee of Guangxi Academy (CGA-00927); the protocol complied with the standard code for the care and use of laboratory animals in China. This research project did not involve endangered or protected species.

### Sample collection and DNA extraction

The adult snails, juvenile snails, and sediment samples were collected from the rice snail breeding demonstration base in Ligao Village (23.37°N, 111.29°E), Liujiang District, Liuzhou City, Guangxi, China. The snails were starved for 24 h before dissection to minimize partially digested food in the intestine (*Van Horn et al., 2012*). Then 30 healthy and undamaged juvenile (3 months old, shell height 28.76 ± 0.44 mm, weight 5.78 ± 0.27 g) and adult snails (1 year old, shell height 43.98 ± 0.91 mm, weight 17.26 ± 0.86 g) were randomly selected from the snails collected in the same habitat (with three biological replicates in each group, and 10 snails in each replicate). Under aseptic conditions, the shells were wiped with 75% ethanol before being removed from each snail and then rinsed with sterile water three times. Each snail was dissected, using sterile tools, on ice in a sterile petri dish. First, the intestines were separated and excess intestinal contents were removed by washing with sterile water three times, then they were homogenized using a Tissuelyser-LT (QIAGEN, Shanghai, China) in a sterile centrifuge tube. In addition, farmland sediment (within 5 cm of the surface mud) was collected from the rice snail system, and then immediately placed into 100 ml sterile frozen tubes and flash frozen with liquid nitrogen. Genomic DNA from all samples was extracted using the HiPure Soil DNA kit (Magen, Guangzhou, China) following the manufacturer's protocol. All extracted DNA samples were stored at −80 °C prior to library construction. In order to avoid the influence of individual differences in snails, this study took the same amount of DNA samples from 10 individuals in the parallel group and pooled them as sequencing samples.

### Library construction and sequencing

The V3–V4 region of 16s rRNA gene was amplified by PCR using universal primers 341F: CCTACGGGNGGCWGCAG and 806R: GGACTACHVGGGTATCTAAT (*Guo et al., 2017*). PCR amplification was performed using high-fidelity KOD polymerase (NEB, Ipswich, UK). PCR reactions were performed in triplicate using 50-μL mixtures containing 5 μL of 10× KOD buffer, 5 μL of 2 mM dNTPs, 3 μL of 25 mM MgSO$_4$, 1.5 μL of each primer (10 μM), 1 μL of KOD polymerase, and 100 ng of template DNA. PCR reagents were obtained from TOYOBO, Japan. PCR conditions were 94 °C for 2 min, followed by 30 cycles at 98 °C for 10 s, 62 °C for 30 s, and 68 °C for 30 s and a final extension at 68 °C for 5 min. The amplicons were pooled, purified, and then quantified using the QuantiFluor™ fluorometer (Promega, Beijing, China). Finally, the amplicons were sequenced using the paired-end strategy (PE250) on the Illumina HiSeq 2500 platform, following standard protocols. The raw reads were deposited into the NCBI Sequence Read Archive database (PRJNA778015).

## Quality control and read assembly

To obtain high-quality clean reads, raw reads containing more than 10% of unknown nucleotides (N), or containing less than 50% of bases with a quality (Q-value) >20, were removed using FASTP (*Chen et al., 2018a*). Paired-end clean reads were merged as raw tags using FLASH (V.1.2.11) with a minimum overlap of 10 bp and mismatch error rate of 2% (*Salzberg, 2011*). Noisy sequences of raw tags were filtered by the QIIME (V.1.9.1) pipeline under specific filtering conditions to obtain high-quality clean tags (*Caporaso et al., 2010*). Briefly, raw tags were broken from the first low-quality base site where the number of bases in the continuous low-quality value (the default quality threshold is ≤ 3) reaches the set length (the default length is 3), and filter tags whose continuous high-quality base length is less than 75% of the tag length. Then, clean tags were searched against the reference database (http://drive5.com/uchime/uchime_download.html) to perform reference-based chimaera checking using the UCHIME algorithm (*Knight, 2011*). All chimaeric tags were removed to obtain effective tags for further analysis.

## Statistical analysis

The effective tags were clustered into operational taxonomic units (OTUs) of ≥97% similarity using the UPARSE (version 9.2.64) pipeline (*Edgar, 2013*). The tag sequence with the highest abundance was selected as the representative sequence within each cluster. Between groups, Venn analysis was performed using the R project Venn Diagram package (version 1.6.16) and an upset plot was performed in the R project UpSetR package (version 1.3.3) to identify unique and common OTUs (*Chen & Boutros, 2011*; *Conway et al., 2017*). The representative sequences were classified into organisms by a naive Bayesian model using RDP classifier (V.2.2) based on the SILVA database, with a confidence threshold value of 0.8 (*Elmar et al., 2007*). The abundance statistics of each taxonomy were visualized using Krona (version 2.6) (*Ondov, Bergman & Phillippy, 2011*). The stacked bar plot of the community composition was visualized in the R project ggplot2 package (version 2.2.1) (*Wickham & Chang, 2008*). A ternary plot of species abundance was plotted using the R ggtern package (version 3.1.0) (*Hamilton & Ferry, 2018*).

For α-diversity analysis, Chao1, Simpson, and all other α-diversity indices were calculated using QIIME (*Caporaso et al., 2010*). An OTU rarefaction curve and rank abundance curves were plotted using the R project ggplot2 package (version 2.2.1) (*Wickham & Chang, 2008*). The α-index comparison between groups was calculated using Welch's t-test in the R project Vegan package (version 2.5.3) (*Neogi et al., 2011*). Differences in α-index among the three groups were assessed with the Kruskal–Wallis H test and Tukey HSD test. For β-diversity analysis, principal coordinates analysis (PCoA) of the Bray–Curtis distances was generated in the R project Vegan package (*Neogi et al., 2011*). The Adonis (also called Permanova) test in the Vegan R package was employed for statistical comparisons of β-diversity among groups. For indicator species analysis, species comparisons between groups was performed using Welch's t-test in the R project Vegan package (version 2.5.3) (*Neogi et al., 2011*). Biomarker features in each group were screened by the labdsv package (version2.0-1) in R project (*Roberts, 2016*). KEGG pathway analysis of the OTUs was inferred using PICRUSt (version 2.1.4) (*Langille et al., 2013*).

## RESULTS

### Bacterial complexity in the sediment and gut microbiome

To study species diversity among the samples, a total of 1,079,092 effective tags were obtained from all samples using Uparse software, and a total of 46,011 valid OTUs were obtained with 97% identity. Evaluating the coverage of the sequencing across all taxa, we found that the rareaction curve tended to asymptote (Fig. S1), which indicated that the sequencing depth covered most of the species richness in the sample.

When analyzed by group, it was found that the number of OTUs in the sediment sample (5,901.2 ± 182.9) was significantly higher than that in the juvenile (2,606.0 ± 121.7) and adult (2,895.7 ± 171.3) snail intestine samples. Among the 6,233 OTUs, 1,452 (23.3%) were shared by three groups, and 434 (7.0%), 355 (5.7%), and 2,785 (44.7%) were unique in the adult and juvenile snail guts, and the sediment samples, respectively (Fig. 1A).

The α-diversity analysis results differed between the three groups. In general, the bacterial diversity in the sediment samples was significantly higher than that in the snail guts ($P < 0.001$), and there was no significant difference in the diversity of the gut flora of adult and juvenile snails ($P > 0.05$), as assessed by the Sobs, Shannon, Chao, and ACE indices (Fig. 1B). This result indicated that the diversity of the gut flora of adult and juvenile snails was similar, but differed from the flora in the sediment.

### Similarity between the sediment and gut microbiomes

PCoA analysis based on the Bray–Curtis distance revealed that the intergroup distance was higher than the intragroup distance (Fig. 1C). In particular, the distance between the gut and sediment samples was the furthest. The Adonis results also showed that the intragroup and intergroup similarity differed for each sample, and the microbial composition of the gut and sediment was different ($R^2 = 0.7923$, $P = 0.003$, Fig. 1C).

### Taxonomic composition of the sediment and gut microbiomes

Among the 6,233 OTUs, the phylum and genus levels accounted for 98.2% and 51.8%, respectively. A total of 47 phyla and 644 genera were identified from all samples in this study.

Proteobacteria and Verrucomicrobia were the two dominant phyla in all samples, with overall relative abundances of 48.2% and 14.2%, respectively (Fig. 2A). The overall abundance of Proteobacteria in the gut of snails (57.0%) was higher than that in the sediment (30.9%), while the overall abundance of Verrucomicrobia in the gut (13.5%) was lower than that in the sediment (15.6%, Fig. 2A). In addition, the relative abundance of Proteobacteria and Verrucomicrobia in the gut of juvenile snails (58.8%, 14.0%) was higher than that of adult snails (55.1%, 13.0%, Fig. 2A). At the genus level, *Aeromonas* dominated (overall abundance: 29.2%) in the gut of snails, followed by *Luteolibacter* (11.2%, Fig. 2B). However, in sediment, *Luteolibacter* was the dominant flora (12.0%), while the abundance of *Aeromonas* was only 0.7%, which differed significantly from the gut ($P < 0.01$, Fig. 2B). Moreover, the relative abundances of *Aeromonas* and *Luteolibacter* in the gut of juvenile snails (30.8%, 11.8%) were also higher than those of adults (27.7%,

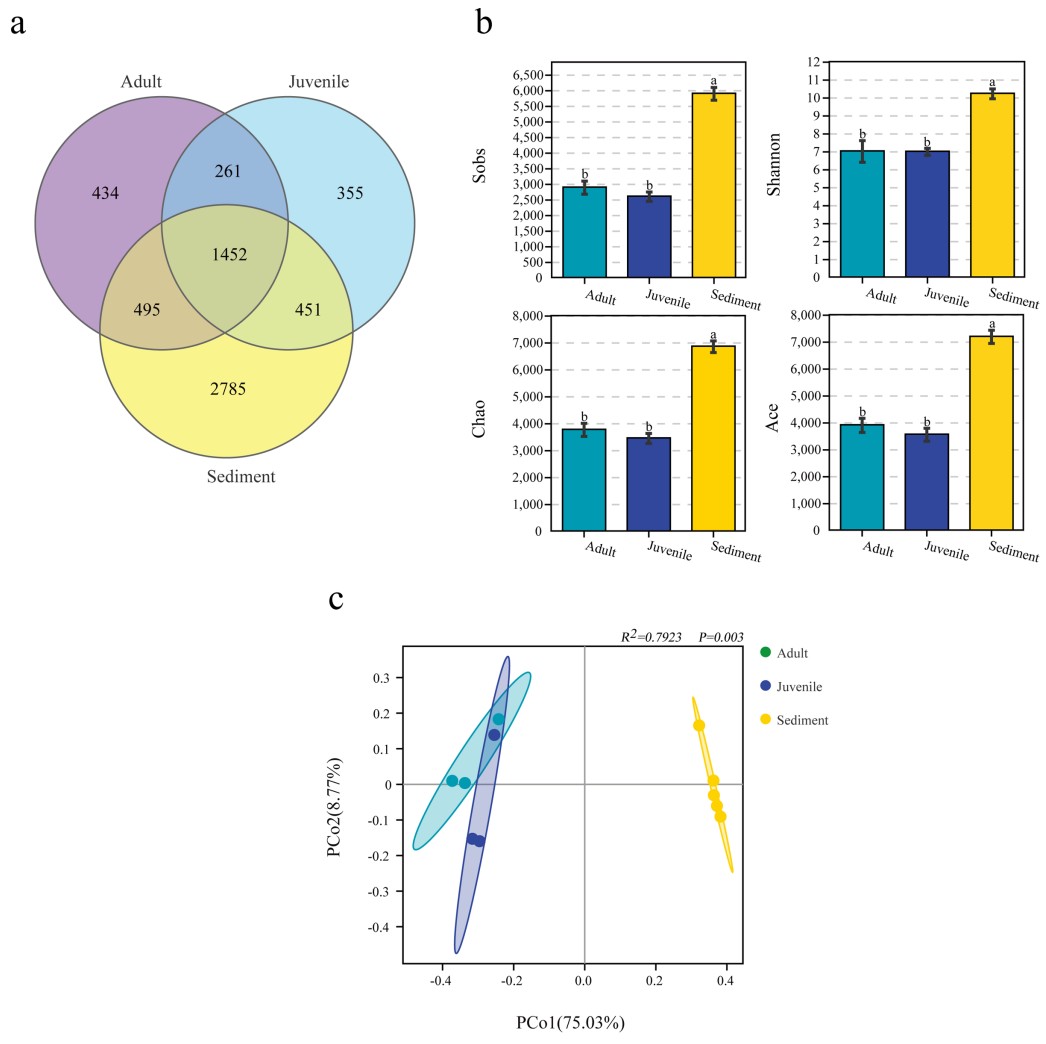

**Figure 1  Number of OTUs (A), α diversity (B) and β diversity analysis (C) of microbiome from gut of *Cipangopaludina chinensis* and sediment.** OTUs: operational taxonomic units. [a], [b]; the group of different letters indicate significant differences ( $P < 0.01$ ).

10.6%, Fig. 2B). Hence, we speculate that *Aeromonas* in the gut of snails may originate from the water environment rather than from the sediment.

## Indicating species of the gut of adult and juvenile *C. chinensis* snails

Welch's t-test was used to analyze differences in the gut microbial composition between adult and juvenile snails, as well as between gut and sediment samples (genus level, filtering the species whose sum of abundance was less than 0.1% in all samples). As shown in Fig. 3B, *Aeromonas* was the most abundant genus in the gut of snails, followed by *Cetobacterium*, *Pseudomonas*, and *Bacteroides*, which were all significantly more abundant than in the sediment ($P < 0.05$). Conversely, *Dechloromonas*, *Sh765B-TzT-35*, *Defluviicoccus*, *SH-PL14*, and *ADurbBin063-1* were all significantly less abundant in the snail gut than in the sediment. Four indicator genera were found, namely *Flavobacterium*, *Silanimonas*, *Geobacter*, and *Zavarzinella*, which were significantly more abundant in the

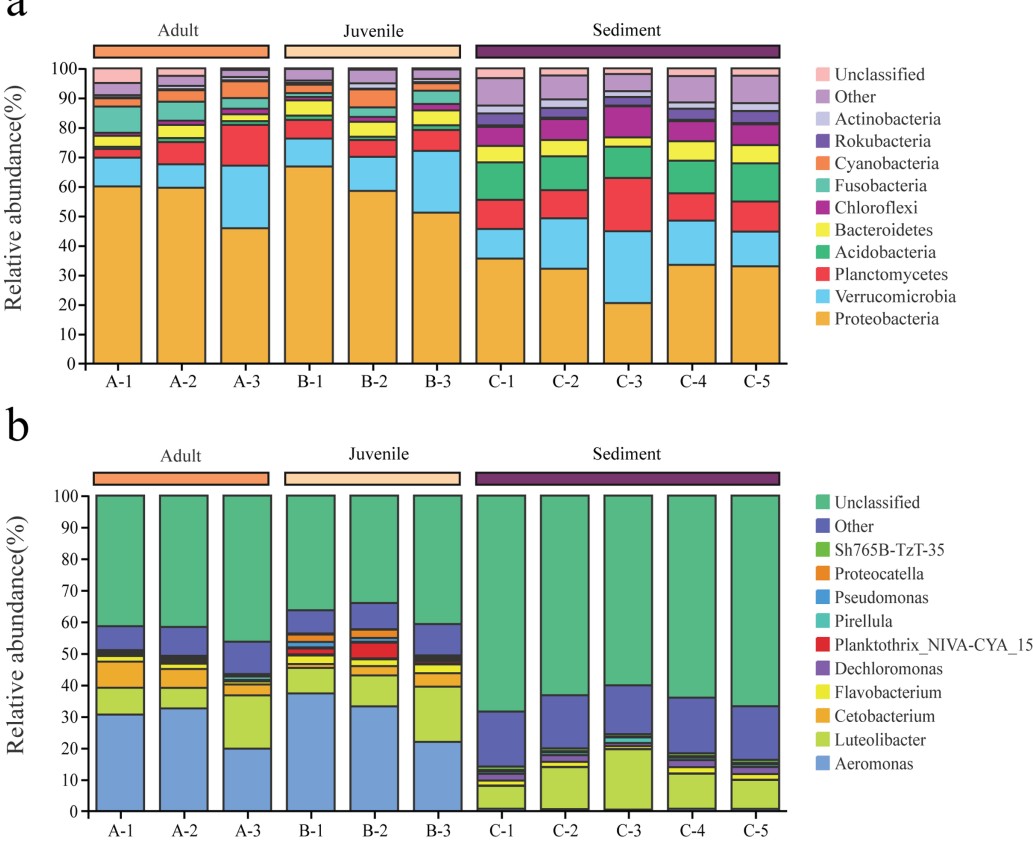

**Figure 2 Composition of the bacterial community in *Cipangopaludina chinensis* gut and sediment at the phylum level (A); at the genus level (B).** A1–A3: adult snails gut samples. B1–B3: juvenile snails gut samples. C1–C5: sediment samples.

gut of juvenile snails than in the gut of adult snails ($P < 0.05$, Fig. 3A). This suggested that these genera were more active at an early age.

## Functional prediction of the sediment and gut microbiomes

The function prediction software PICRUSt was used for analysis (Fig. 4). The results showed that the abundance of microbes of all functions was significantly lower in the sediment than in the gut ($P < 0.05$). Representative functions included metabolism of cofactors and vitamins, amino acid metabolism, carbohydrate metabolism, and fatty acid metabolism. There was no statistical difference in the gut microbial function between adult and juvenile snails ($P > 0.05$), which implied that the nutrients required during the growth and development of snails were consistent.

## DISCUSSION

In snails, the gut is the main site for nutrient absorption and utilization (*Pawar et al., 2012*; *Dar, Pawar & Pandit, 2017*). Currently, little is known about the structure and function of the gut flora of *C. chinensis*. Previously, we found that the utilization of protein and carbohydrates by juvenile snails was higher than that of adult snails (*Zhou et al., 2021*). There are conflicting reports regarding changes to the microbiota with age, with some

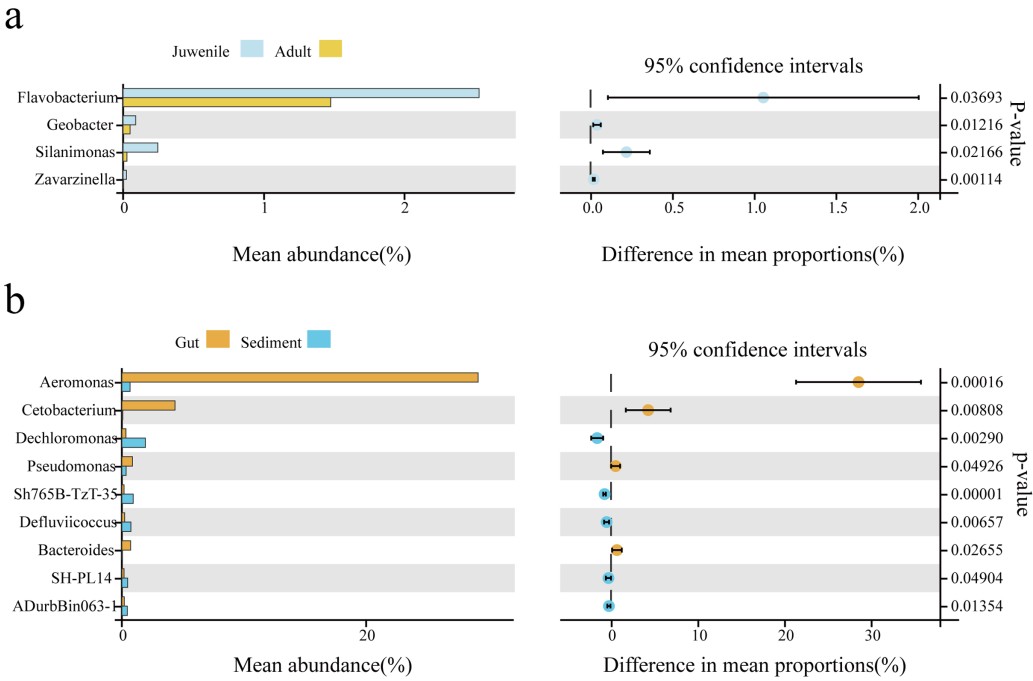

**Figure 3 The biomarker features in *Cipangopaludina chinensis* gut and sediment at the genus level using Welch's t-test.** (A) An analysis of gut biomarker features of the adult and juvenile snails. (B) An analysis of the biomarker features between the sediment and the snail gut. The histogram shows the relative abundance of different species in the two groups; the coordinates of the point right figure were the abundance difference, and the error bar shows the fluctuation range of the difference in the 95% confidence interval, and the *P* value on the far right.

studies reporting that microbial communities differ from birth to adulthood, and others showing relatively consistent gut microbiota in adult and juvenile animals (*Stephens et al., 2016*; *Xue et al., 2015*). Our research found no statistical difference in the number of OTUs or α-diversity between adult and juvenile snails. Our PCoA results confirmed clustering of these two groups, which indicated that the intestinal flora of *C. chinensis* was relatively stable during the growth process.

The colonization of the gut flora of aquatic animals is a complex process affected by many factors such as the sediment, water environment, and bait (*Romero & Navarrete, 2006*). Previous studies have found that *C. chinensis*, a type of bottom-breathing organism, contains a lot of humus and sediment in the gut (*Zhou, 1986*). To date, few studies have analyzed the difference in microbiota composition between aquatic animal guts and the sediment in an ecosystem. In our research, the number of OTUs (5,901.2) in the sediment was about twice that of the *C. chinensis* gut (mean 2,750.9), and the microbial diversity of the sediment was significantly different to that of the snail gut. In aquaculture, the sediment is in an open environment, rich in organic matter and microorganisms, and plays an important role in supplying fertilizer and regulating water quality (*Gilbride et al., 2006*). The diversity of bacterial communities has been reported to contribute to biochemical reactions within the ecosystems of sediments (*Gilbride et al., 2006*). By contrast, the gut of a snail is a relatively closed environment, and the richness and diversity of the microbial

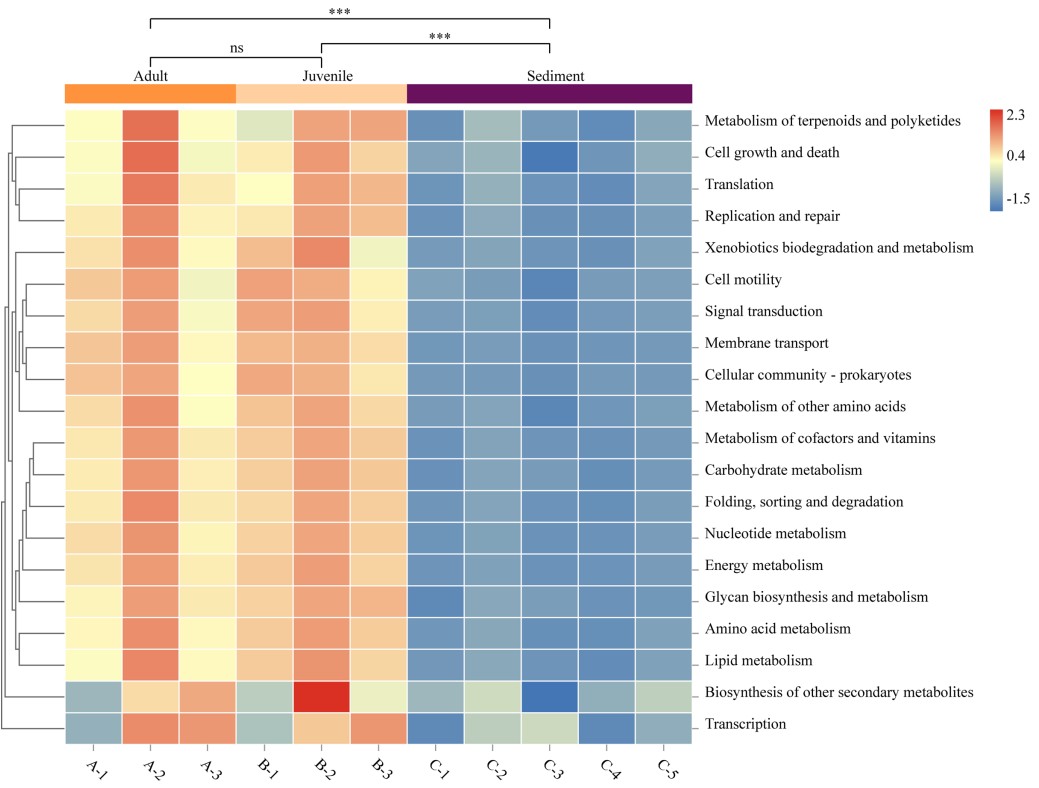

**Figure 4 Functional prediction of sediment and *Cipangopaludina chinensis* gut microbial communities using PICRUSt.** Asterisks (***) represent an extremely significant difference ($P < 0.01$). ns: indicates no significant difference ($P > 0.05$).

community is predominantly affected by food intake, water quality, and sediment, which was consistent with our findings.

In our study, we found the gut microbiota composition of adult and juvenile snails to be similar, with the main phyla being Proteobacteria and Verrucomicrobia, and the main genera being *Aeromonas* and *Luteolibacter*. In addition to Proteobacteria, the gut microbiotas of vertebrates have previously been reported to be enriched in Firmicutes, Bacteroidetes, and Fusobacteria. In the gut of *Radix auricularia* snails, Mycoplasmataceae and Chloroflexaceae were the dominant bacteria (*Hu et al., 2018*), and *Aeromonadaceae*, *Sediminibacterium*, and *Cloacibacterium* were the most abundant genera in the gut of *Pomacea canaliculata* snails (*Li et al., 2019*). The composition of the intestinal microbial communities reflects natural selection between the host and microorganism, thereby promoting the functional stability of the intestinal microecosystem (*O'Hara & Shanahan, 2006*). The different compositions of the gut microbiotas of different aquatic species may be attributed to differences in habitat, season, and genetic characteristics (*Nicolai et al., 2015*).

In our study, Proteobacteria as identified as one of the main bacterial phyla in sediment, which was consistent with a previous report that Proteobacteria was the most abundant flora in rice field sediment (*Chen et al., 2012*; *Zhao et al., 2017*). At the genus level, we found that the abundance of *Aeromonas* in the gut was significantly higher than that in the

sediment. *Aeromonas* is ubiquitous in aquatic environments (*e.g.*, water, food, and sediment), and the detection rate in shellfish aquatic products reaches 61.29% (*Fei, 2017*). Recently, healthy animals have also been found to carry this bacterium along with changes in habitat. One study found that *Aeromonas* was the main intestinal bacterium in the microbiotas of three planorbid snails (*Bulinus africanus*, *Biomphalaria pfeifferi*, and *Helisoma duryi*) (*Van Horn et al., 2012*). Similar results were also observed in the intestines of *P. canaliculata* snails. Furthermore, *Hu et al. (2018)* suggested that *Aeromonas* may play a prominent role in the degradation of cellulose in *R. auricularia* snails. We also detected the presence of *Aeromonas* in the gut of healthy snails in this study, speculating that this potentially cellulose-degrading bacterium is common in aquatic animals.

To explore the potential indicator flora of the adult and juvenile snails gut, indicator species analysis was performed using the OmicShare tools based on 16rRNA data. The abundance of four index species (*Flavobacterium*, *Silanimonas*, *Geobacter*, and *Zavarzinella*) was significantly higher in the gut of juvenile snails compared with adult snails. *Flavobacterium* is a Gram-negative bacillus widely found in sediment and water (*Chen et al., 2018b*; *Kim & Yu, 2020*). *Geobacter* is an important $Fe^{3+}$-reducing dissimilating bacterium, which has a significant impact on the community structure of iron-reducing dissimilating microorganisms in paddy field soil (*Chen et al., 2019*). Therefore, *Flavobacterium* and *Geobacter* in the snail gut likely originate from the sediment. Compared with adult snails, juvenile snails are small caliber with weak feeding ability and are therefore unable to eat large phytoplankton. Instead, they ingest a large amount of humus from the sediment to provide nutrition. It was a surprising result that *Silanimonas* was one of the indicator species for the two growth stages of snails, suggesting that this bacterium may be an important source of nutrients in the early development of snails. Hence, *Silanimonas* has the potential to be developed into open bait for the early growth of snails.

Our findings also revealed that the phyla Proteobacteria, Fusobacteria, and Tenericutes were significantly more abundant in the snail gut than in sediment. Previous studies have also shown that these phyla are predominant in the gut of aquatic animals (*Fei, 2017*). In particular, Proteobacteria has been reported to be a microbial indicator of a gut flora imbalance (*Shin, Whon & Bae, 2015*). This implied that the gut microenvironment of the snail is in a relatively stable state. Interestingly, we found cyanobacteria in the gut of the snail, a bacterium known to be widely distributed throughout aquatic environments. This finding confirmed that *C. chinensis* can ingest cyanobacteria, which provides possibilities for water purification and bloom control in the future. We also detected the unique microbial communities present in the sediments of paddy fields (*e.g.*, Acidobacteria, Actinobacteria, Chloroflexi, and Nitrospirae). *Lin et al. (2020)* used high-throughput sequencing to identify Nitrospirae and Acidobacteria as the dominant flora in paddy fields, and *Singh et al. (2019)* reported that Nitrospirae, Actinobacteria, and Acidobacteria were widespread in acidic water systems. Therefore, we suspect that the residual water-soaked rice stalks in paddy fields produce acidic substances through microbial fermentation, which lowers the pH of the water, producing an acidic water system, which promotes the colonization and development of these acidophilic bacteria.

We explored the functional differences between the bacterial communities in snail guts and sediment using PICRUSt. As shown in Fig. 4, the functional classification of microorganisms in the gut and sediments were similar, but there were differences at the level of gene expression. In particular, microorganisms involved in metabolism of cofactors and vitamins, amino acid metabolism, carbohydrate metabolism, and lipid metabolism were significantly more abundant in the gut microbiota than in the sediment. Studies have confirmed that *C. chinensis* snails mainly feed on algae (green alga, cyanobacteria, *Silanimonas*) (*Zhou, 1986*). To degrade plant fiber, the expression of specific functional genes in the gut flora may be needed, such as genes associated with fatty acids, amino acids, vitamins and cofactors (*Cardoso et al., 2012*). This may be one of the reasons for the functional differences in microbial gene expression in the snail gut compared with the sediment. There was no difference in the function of the gut microbes between adult and juvenile snails, indicating that there is no significant change in the nutrient requirements or composition required during the growth of *C. chinensis*.

## CONCLUSION

The development of sequencing technology has provided a new way to study the microbial communities of lower mollusks. Our work explored the microbiotas of adult and juvenile snail guts, as well as comparing the microbiotas between snail guts and sediment in the same habitat. Our findings revealed that the microbial profiles of snail guts and sediments differed, but their microbial communities were closely related, indicating that changes in the composition of snail gut microbes were tightly associated with the sediment in the same ecosystem. This provides guidance for future studies on the interaction between the gut flora of snails and their environment. The growth and development of the snails did not greatly affect the composition of the gut flora, and the functions of the gut flora at different developmental stages were similar, suggesting that the gut microecological environment of the snails was relatively stable. This study found that *Silanimonas* may be used as an open food for juvenile snails in culture. Our findings provide valuable insight into the relationship between snails and environmental microorganisms.

### Funding

This study was supported by the Guangxi Natural Science Foundation (2020GXNSFBA297067), the Guangxi Science and Technology Base and Talent Project (AD21220010), the National Modern Agricultural Technology System (nycytxgxcxtd-2021-08) and the Guangxi Innovation Driven Development Special Fund (AA20302019-5). The funders had no role in study design, data collection and analysis, decision to publish, or preparation of the manuscript.

### Grant Disclosures

The following grant information was disclosed by the authors:
Guangxi Natural Science Foundation: 2020GXNSFBA297067.

Guangxi Science and Technology Base and Talent Project: AD21220010.
The National Modern Agricultural Technology System: nycytxgxcxtd-2021-08.
Guangxi Innovation Driven Development Special Fund: AA20302019-5.

## Competing Interests

The authors declare that they have no competing interests.

## Author Contributions

- Kangqi Zhou conceived and designed the experiments, prepared figures and/or tables, authored or reviewed drafts of the paper, and approved the final draft.
- Junqi Qin conceived and designed the experiments, prepared figures and/or tables, authored or reviewed drafts of the paper, and approved the final draft.
- Haifeng Pang performed the experiments, prepared figures and/or tables, authored or reviewed drafts of the paper, and approved the final draft.
- Zhong Chen analyzed the data, prepared figures and/or tables, and approved the final draft.
- Yin Huang performed the experiments, prepared figures and/or tables, and approved the final draft.
- Wenhong Li analyzed the data, prepared figures and/or tables, and approved the final draft.
- Xuesong Du analyzed the data, prepared figures and/or tables, and approved the final draft.
- Luting Wen performed the experiments, prepared figures and/or tables, and approved the final draft.
- Xianhui Pan conceived and designed the experiments, authored or reviewed drafts of the paper, and approved the final draft.
- Yong Lin conceived and designed the experiments, authored or reviewed drafts of the paper, and approved the final draft.

## Data Availability

The sequences are available at SRA: PRJNA778015.

## Supplemental Information

Supplemental information for this article can be found online at http://dx.doi.org/10.7717/peerj.13042#supplemental-information.

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
