# Peer review of "Comparison of the composition and function of gut microbes between adult and juvenile Cipangopaludina chinensis in the rice snail system"

_PeerJ, doi:10.7717/peerj.13042_

## Round 0.1 · original submission · Major Revisions

There are serious issues regarding sampling and sample size; please address all the raised issues by the honorable reviewers before resubmitting.

Reviewer 1 ·

Basic reporting

Manuscript :
Comparison of the composition and function of gut microbes between adult and juvenile Cipangopaludina chinensis in the rice snail system

The manuscript described the differences in gut microbiota complexity in Juvenile, adult, and sediment of Cipangopaludina chinensis. The work is novel as characterization of mollusk microbiota did not clarify or extensively studied as other aquatic fish species. I recommend the acceptance of the manuscript but after minor revision as shown below;
In the introduction more detailed about the anti-cancer effects of Cipangopaludina chinensis could increase the scientific value of the manuscript.
The authors stated that”Therefore, C. chinensis can not only be used as food, but also have great potential in human disease prevention and treatment” In relation to this more information about the nutritional values and ingredients of this mollusk is needed. Additionally, the hepatoprotective effects should be mentioned.
In the discussion the authors should highlight the abundance of healthy bacteria (if possible) among the identified gut microbiota either in Juvenile or adults.
Results are clearly presented and the references are updated

Experimental design

The experimental work is well designed, clear, and reproducible

Validity of the findings

Up to the economical problems especially in third world and poor countries working on snails as source of nutritional proteins could be of great help. Based on the importance of gut-organ axis and its relationship to health and disease, characterization of C. chinensis microbiota could help the industrial contribution to increase the abundance of C chinensis probiotics as nutritional intervention.

Additional comments

The manuscript is well written and the interpretation is satisfactory

Reviewer 2 ·

Basic reporting

The manuscript is good written. For further improvement, few suggestions are recommended for revision as follows and as shown in the annotated manuscript.

Experimental design

1. The introduction section needs more details regarding feasibility of the sequencing method for this study, especially why 16SrRNA is used.
2. In lines, 53-55 and 60-61, rephrasing is required with correct language expression.
3. In line 99, “Bioinformatics and community comparisons” should be expanded.

Validity of the findings

In the reference section (lines, 303-305), names of journal should be appropriate as they are shown originally.

Additional comments

N/A

Reviewer 3 ·

Basic reporting

I have specific suggestions about the language, but there are more important points addressed below.

Experimental design

ok

Validity of the findings

I believe the work is important. But to strengthen the work, I would suggest that the authors would have to replicate it at least three times in different locations. Additionally, it says 30 samples were collected per age, but when you look at the data you only see three samples (data points). Then it says that there was a random selection of samples. I could not understand if the random selection was from the original environment or after collecting the sample, some were randomly selected for sequencing. But if a fraction of samples were selected for sequence then all of that should have been explained explicitly and I could (maybe I am mistaken) retrieve that information with clarity from the materials and methods or legends. Besides, if 30 samples were collected, why were not all sequenced and analyzed? Were they pooled? As for data visualization of taxa, I would also suggest not showing the data for individual samples but instead plotting collectively with box and whiskers or some other ways to show the variation across host and environment. Another consideration would be to confirm the observed differences by 16S rRNA sequencing using qPCR for a specific genus (e.g., Aeromonas). Also, I would suggest changing the coloring scheme for all plots with the exception of Figures 3 and 4.

Additional comments

I emphasize that I empathize with the importance of the work. But I respectfully believe there is extra work to be done prior to publication.

---

## Round 0.2 · Major Revisions

Still there is major concern regarding stated objective in relation to the findings. I request you to address the issue properly raised by the honorable reviewer.

Reviewer 3 ·

Basic reporting

ok

Experimental design

In the rebuttal the authors said "Moreover, we extracted DNA for each parallel group, and then extracted an equal amount of DNA samples from the 10 individuals in the parallel group and pooled them for library construction, in order to avoid the influence of individual differences."
My previous review was to reject the paper based on the sample size obscurity. Now that is clarified. But if you examine the initial study objective as stated in the intro:

"Herein, it is necessary to systematically understand the dynamic
67 changes of intestinal microflora of C. chinensis so as to develop the best diet for snails. "

"In this study, we performed high-
71 throughput sequencing of 16S rRNA gene to study the function and composition of intestinal microbiota in
72 adult and juvenile C. chinensis under artificial habitat. This result provides insight into the reasons for
73 differences in feeding behavior and food preferences between juvenile and adult snails. "

In my view, individual responses would have been necessary to achieve this goal. If it were just a survey of microbiome compositional differences, I presume pooling would be fine. But given the initial objective, my opinion remains the same. The study is valid, and the other changes are all acceptable and great. I just think that as the objectives are stated it cannot be accepted. Now if this is published a survey of microbiome differences as a hypothesis-generating study, then my opinion would be to accept it.

Validity of the findings

NA

Additional comments

NA

---

## Round 0.3 · accepted · Accept

Congratulations!

Finally the reviewers' are satisfied with the revision! Hope your work will contribute to the knowledge enhancement of the gut microbiome study of non-human animals in a particular ecosystem.

Reviewer 3 ·

Basic reporting

ok

Experimental design

ok

Validity of the findings

ok

Additional comments

ok

---

## Author Rebuttal · Round 0.3

Dear Editors

My manuscript **"Comparison of the composition and function of gut microbes between adult and juvenile *Cipangopaludina chinensis* in the rice snail system"** has been revised according to your guidance. We thank the reviewers for their generous comments on the manuscript and have have edited the manuscript to address their concerns. We hope that they will be satisfied.

Under your guidance, we believe that the manuscript is now suitable for publication in PeerJ.

Sincerely yours

Mr. Xianhui Pan

On behalf of all authors

Reviewer 3 (Anonymous)

Basic reporting

Ok

**Reply**:Thanks for the affirmation of the reviewers.

Experimental design

In the rebuttal the authors said "Moreover, we extracted DNA for each parallel group, and then extracted an equal amount of DNA samples from the 10 individuals in the parallel group and pooled them for library construction, in order to avoid the influence of individual differences."

My previous review was to reject the paper based on the sample size obscurity. Now that is clarified. But if you examine the initial study objective as stated in the intro:

"Herein, it is necessary to systematically understand the dynamic

67 changes of intestinal microflora of C. chinensis so as to develop the best diet for snails. "

"In this study, we performed high-

71 throughput sequencing of 16S rRNA gene to study the function and composition of intestinal microbiota in

72 adult and juvenile C. chinensis under artificial habitat. This result provides insight into the reasons for

73 differences in feeding behavior and food preferences between juvenile and adult snails. "

In my view, individual responses would have been necessary to achieve this goal. If it were just a survey of microbiome compositional differences, I presume pooling would be fine. But given the initial objective, my opinion remains the same. The study is valid, and the other changes are all acceptable and great. I just think that as the objectives are stated it cannot be accepted. Now if this is published a survey of microbiome differences as a hypothesis-generating study, then my opinion would be to accept it.

**Reply**:Thank you for your professional and constructive comments. The purpose of this study was to preliminarily explore the differences in gut microbial communities in different growth stages of C. chinensis (adult and juvenile stage). We have revised and refined lines 72-73 and 80-82 in the manuscript based on the comments of the reviewers, which hopefully meet the reviewers' requirements.